# Single Mutation in Hammerhead Ribozyme Favors Cleavage Activity with Manganese over Magnesium

**DOI:** 10.3390/ncrna6010014

**Published:** 2020-03-20

**Authors:** Mohammad Reza Naghdi, Emilie Boutet, Clarisse Mucha, Jonathan Ouellet, Jonathan Perreault

**Affiliations:** 1Institut National de la Recherche Scientifique (INRS), Centre Armand Frappier Santé Biotechnologie, 531 boul. des Prairies, Laval, QB H7V 1B7, Canada; MohammadReza.Naghdi@iaf.inrs.ca (M.R.N.); Emilie.Boutet@iaf.inrs.ca (E.B.); clarisse.mucha@iaf.inrs.ca (C.M.); 2Department of Chemistry and Physics, Monmouth University, 400 Cedar Avenue, West Long Branch, NJ 07764, USA; jouellet@monmouth.edu

**Keywords:** manganese, magnesium, cation, RNA variants, C6 hammerhead variants, A6

## Abstract

Hammerhead ribozymes are one of the most studied classes of ribozymes so far, from both the structural and biochemical point of views. The activity of most hammerhead ribozymes is cation-dependent. Mg^2+^ is one of the most abundant divalent cations in the cell and therefore plays a major role in cleavage activity for most hammerhead ribozymes. Besides Mg^2+^, cleavage can also occur in the presence of other cations such as Mn^2+^. The catalytic core of hammerhead ribozymes is highly conserved, which could contribute to a preference of hammerhead ribozymes toward certain cations. Here, we show a naturally occurring variation in the catalytic core of hammerhead ribozymes, A6C, that can favor one metallic ion, Mn^2+^, over several other cations.

## 1. Introduction

Independent discoveries by the laboratories of Thomas Cech and Sidney Altman, leading to a shared Nobel prize in chemistry in 1989, demonstrated that RNA could catalyze chemical reactions [1,2,3,4,5]. Naturally occurring ribozymes [3,4,5,6] include group I [1] and II [7] introns, RNAse P RNA [2], spliceosomal RNA [8] and ribosomal RNA [6], as well as small hammerhead [9,10], Varkud satellite (VS) [11], hairpin [12], hepatitis delta virus (HDV) [13] (and HDV-like [14]), twister [15] (and twister sister), pistol, hatchet [16] and *glmS* ribozymes [17]. Hammerhead ribozymes (HHRz) were observed for the first time in the late eighties in tobacco plants [9,10]. The HHRz were the most studied ribozymes for self-cleavage activity, becoming models for research on RNA structure and function [18]. Since then, it has been shown that HHRz are widespread and could be found in all domains of life [19,20,21,22].

At physiological pH level, the activity of HHRz depends on metal ions, especially Mg^2+^ [23], which supports cleavage in vitro for a minimal, but sub-optimal, HHRz sequence at 10 mM [24]. Other ions can also activate the self-cleavage of HHRz [25]: cations like ammonium (NH_4_^+^) can support the activity of HHRz [26] and large tetraalkylammonium ions significantly increase the rate of HHRz in addition to Mg^2+^ [27]. The cleavage rate of HHRz was tested with transition metals and depending on the conditions and ribozymes tested, cleavage with Mn^2+^ showed three times [28] and up to seventy-six times [29] better cleavage than Mg^2+^. In fact, metal ions like Mn^2+^ bind to specific nucleotides of the catalytic core, such as the phosphate of A9, the nitrogen from G10.1 and the oxygen of G12 [30,31,32,33] (Figure 1). Nevertheless, the finding that Mn^2+^ bound to hammerhead ribozymes and bound more strongly than Mg^2+^ or K^+^ [34] is not surprising given that Mn^2+^ also binds RNA better, in general [35].

The minimal catalytic core of HHRz is made of the core consensus C3U4G5A6NG8A9–G12A13A14 with the A15–U16 base pair and H17 cleavage site surrounded by three helical stems [25] (Figure 1), which are necessary for cleavage activity. Nevertheless, some rare variations at certain core positions decrease the cleavage rate in a few natural HHRz, but the ribozymes presumably remain functionally active in vivo [20].

Two examples of variants, U(2a)G(2b)U(3)U4G5A6C7G8A9 and G(2a)C(2b)C(3)U4G5A6C7G8A9 from halophilic organisms, were suggested to modulate gene expression according to divalent cation concentrations [20]. We hypothesized that some other HHRz would also be likely to have varying ion specificity. We set our goal to determine first whether a previously identified core variant (A6C) from bacteriophage *Bcep176* could have altered cation preferences, and second if this single A6C substitution within the core could alter ion preference for other HHRz. To keep the naming convention clear, the natural variant *Bcep176* will be denoted as *Bcep176* (C6). In this paper, we show that this naturally occurring variation from the typical catalytic core is deleterious for cleavage activity with Mg^2+^ (and other divalent cations), but still allows good cleavage activity with Mn^2+^.

## 2. Results

### 2.1. Varying Metal Ion Preference of a HHRz Variant

We assayed over a dozen putative ribozymes (selected from [20]) that either had a variant core or gene context suggestive of cation regulation (Appendix A). Five were active in our assay conditions, including the *Bcep176* (C6) variant which barely cleaved during transcription, but was active in the presence of Mn^2+^ after purification (Appendix A). We determined how this natural variation (C6) could affect the cleavage of *Bcep176* (C6) in the presence of various ions and we found marked differences between activity in Mg^2+^ and Mn^2+^. To verify the specificity of *Bcep176* (C6) for metal ions, Mg^2+^, Mn^2+^ and other metals such as Ca^2+^, Zn^2+^, Ni^2+^, Co^2+^, Cd^2+^ and Cu^2+^ were tested at 0.01, 0.1 and 1 mM, with the exception of Cu^2+^, which was tested at 0.01 and 0.1 mM (Figure 2A). Cleavage occurred solely in the presence of either Mg^2+^ or Mn^2+^. To determine the cleavage activity of RNA *Bcep176* (C6), assays were performed for up to 60 min in the presence of Mg^2+^ at 0.3, 1, 3 and 10 mM; and for Mn^2+^ at 0.01, 0.03, 0.1, 0.3, 1 and 3 mM (Figure 2B,C). The cleavage activity of *Bcep176* (C6) at 0.1 mM Mn^2+^ was observed, whereas no cleavage activity was observable at 0.1 mM Mg^2+^ (Figure 2B,C). The k_obs_ values were calculated as 0.31 min^−1^ and 0.29 min^−1^ at 1 and 3 mM Mn^2+^, respectively (Figure 2C), whereas k_obs_ values at equivalent Mg^2+^ concentration were calculated as 0.0041 min^−1^ and 0.051 min^−1^, respectively (Figure 2B).

To better understand the role of *Bcep176* (C6) regarding higher activity with Mn^2+^ compared to Mg^2+^, we did the inverse mutation C6A, leading to *Bcep176* (C6A), reverting to consensus A6 and a negative control mutation of GAAA → GUUU (sequences B and C of Figure 3, respectively). Cleavage did not take place in the inactive mutant GAAA → GUUU, as expected. In contrast, for the inverse mutant *Bcep176* (C6A), more efficient cleavage took place, both in the presence of 1 mM Mn^2+^ and during transcription (25 mM Mg^2+^) (Figure 3D), indicating that the “consensus-like” *Bcep176* (C6A) mutant did not discriminate between Mg^2+^ and Mn^2+^. The higher cleavage (36% and 46%) observed in the presence of 0.1 and 1 mM Mn^2+^, respectively, for native *Bcep176* (C6) suggests that the nucleotide C6 in this WT hammerhead causes a preference for Mn^2+^ over Mg^2+^ (Figure 3E). The fact that *Bcep176* (C6) barely cleaved during transcription (25 mM Mg^2+^) (Figure 3D), but cleaved to ~40% with 10 mM Mg^2+^ (Figure 2 and Figure 3E), may be due to differences of folding during in vitro transcription compared to folding after purification and snap cooling.

### 2.2. Effect of A6C Mutation on Another HHRz

Furthermore, we used a different HHRz with high self-cleavage activity in the presence of Mg^2+^ to explore if an A6C mutation within a consensus HHRz catalytic core would lead to the same phenotype, i.e., cleavage in the presence of Mn^2+^ would be favored over cleavage with Mg^2+^. The mutated CUGCUGA version of a pseudoknotted type II HHRz derived from mouse gut (hereinafter referred to as mouse gut HHRz) (from [20]) (Figure 4A) cleaved better with Mn^2+^ compared to Mg^2+^, similar to that observed with *Bcep176* (C6). This native mouse gut HHRz (A6) showed high self-cleavage activity during transcription (25 mM Mg^2+^). The native mouse gut HHRz had a similar cleavage efficiency (k_obs_ = 0.3 min^−1^) with Mn^2+^ and Mg^2+^ at 300 µM (Figure 4B,C, Table 1; Appendix A). However, for the mouse gut HHRz (A6C) mutant, there was a greater than 10,000-fold rate difference between Mn^2+^ and Mg^2+^ at the same ion concentration of 300 µM (k_obs_ = 0.18 min^−1^ vs k_obs_ = 3.55 × 10^−6^ min^−1^, respectively) (Figure 4B,C, Table 1; Appendix A). It should be noted that even if the C6 mutation favors better cleavage with Mn^2+^ over Mg^2+^, our data does not necessarily indicate better binding for Mn^2+^ over Mg^2+^.

## 3. Discussion

The increasing discovery of ncRNAs, especially riboswitches and ribozymes, is in large part due to powerful tools in bioinformatics. As shown previously, these approaches discovered several hammerhead ribozymes such as *Bcep176* (C6) [20]. In this paper, we determined the activity of this previously found HHRz with a variant base within the catalytic core, as well as the A6C equivalent mutant of the mouse gut HHRz. In both the cases, the C6 HHRz core nucleotide resulted in an apparent preference for Mn^2+^ for cleavage activity.

While Mg^2+^ is usually regarded as the most relevant ion for HHRz cleavage activity in physiological conditions, we can imagine that particular structural variants of HHRz could exhibit a preference for other divalent cations such as Mn^2+^. This might be an example of how HHRz could putatively act as divalent cation sensors and suggests that more functions remain to be discovered in the treasure trove of known and unknown ribozymes, as was also previously suggested for some HDV-like ribozymes [36]. The tested concentration of ions could be biologically relevant in some context, knowing that intracellular concentrations of Mn^2+^ can reach several hundred micromolar in some conditions [37]. Nevertheless, we do not know if this is biologically relevant, especially given that concentrations of Mg^2+^ are typically a few orders of magnitude higher than for Mn^2+^. Thus, apart from *glmS*, the biological function of small ribozymes in bacteria remains unclear [38].

Past work with a minimal HHRz showed that an abasic position #6 decreased the cleavage activity in the presence of both the monovalent ion Li^+^ and divalent Mg^2+^ [39]. We, however, demonstrated that a native *Bcep176* (C6) and mutated mouse gut HHRz (A6C) have a change in cation preference compared to the typical A6 core position, greatly reducing cleavage in the presence of Mg^2+^, but barely affecting cleavage with Mn^2+^ (in some conditions) (Appendix A, Table 1). Our results imply that in vitro consensus core mutations can change the cation preference of a hammerhead ribozyme, similar to a change of ligand specificity observed by others for *glmS* ribozymes [40]. This is also similar to past work that showed that the mutation G12A in HHRz can change specificity from Mg^2+^ to Zn^2+^ [41], although in this case the reason for this change is better understood because this nucleotide directly interacts with the metal ion. In the case of C6, the exact molecular interactions responsible for changing the cleavage preference for Mn^2+^ are not known. Among the tested cations, Cd^2+^ has coordination very similar to Mn^2+^ [42], which might thus be expected to affect the folding of ribozyme *Bcep176* (C6) similar to Mn^2+^, but the assays for cleavage activity did not yield similar results, implying that coordination is not the only factor affecting cleavage activity. The fact that such small changes in sequence can lead to a drastically altered ion specificity is a fascinating example of how evolution could potentially and easily alter existing scaffolds to achieve new functions. Five atoms of Mn^2+^, including one which binds to A9, could be observed to bind HHRz with standard consensus (A6) in an atomic resolution structure [31]. Future work regarding Mn^2+^ in the context of the cytidine C6, through determination of atomic resolution structure, for instance, could be informative on how this small change has such an impact on ion preference for cleavage activity.

Several applications can be found for the HHRz with the C6 position mutation. For example, since *Bcep176* (C6) HHRz has good self-cleavage activity under 0.3 mM Mn^2+^, as opposed to a very low activity with Mg^2+^ (Appendix A, Table 1), it might be usable as a Mn^2+^ sensor. For other applications, introducing the mutation A6C to a highly active HHRz to bypass self-cleavage during transcription (high concentration of Mg^2+^) would allow the HHRz to be selected later under desired conditions in the presence of Mn^2+^.

## 4. Materials and Methods

### 4.1. PCR Product of Wild-Type and Mutant Ribozymes

The sequences of bacteriophage *Bcep176* (C6) and mouse gut HHRz were selected and the primers were designed using Primerize [43] for the wild-type and mutated versions, with the addition of the T7 polymerase promoter sequence. The whole sequence was constructed using assembly PCR. In the same way, two constructs having mutations were produced, one to change C6 → A6 to mimic the consensus core of HHRz within the *Bcep176* HHRz and the other to change GAAA → GUUU to create an inactive *Bcep176* HHRz. The A6C mutation in the catalytic core of the mouse gut HHRz from the mouse gut metagenome [20] was also created by assembly PCR. Oligonucleotides used for these PCR assemblies are listed in Table 2.

### 4.2. RNA Transcription

The PCR product of *Bcep176* (C6) and its mutants were subjected to radiolabeling with [α-^32^P] UTP during transcription for three hours by using 10 µL of transcription buffer (5x: 400 mM HEPES-KOH, pH 7.5, 120 mM MgCl_2_, 10 mM spermidine and 200 mM DTT), 15 µL of rNTP (a mix of 10 mM ATP, 10 mM GTP, 10 mM CTP and 0.4 mM UTP), 10 µL of DNA template (~200 ng), 1 µL of pyrophosphatase 50X, 1 µL of RNAse inhibitor (40 U/µL), 0.5 µL of [α-^32^P] UTP and 2 µL of T7 RNA polymerase (at least 25 U/µL). The volume of the reaction was completed with RNAse-free water. The samples were incubated for 2 h and in some cases, for one more hour after adding an additional 1 µL of T7 RNA polymerase to increase the yield. The RNA was precipitated in −20 °C by 0.1 volume of sodium acetate (CH_3_COONa; 3 M, pH 5.2) and 2 volumes of 100% cold ethanol for 2 h.

The precipitated uncleaved RNA was purified by migration on 8% denaturing polyacrylamide gel (8 M urea PAGE, polyacrylamide gel electrophoresis) and eluted in elution buffer (0.3 M NaCl) for two hours at room temperature. The eluent was precipitated as mentioned previously.

The mouse gut HHRz (A6) exhibited high self-cleavage activity in the presence of Mg^2+^, so it cleaved at high levels during transcription, impeding our capacity to select the uncleaved part of the ribozyme for further analysis. Therefore, we added a complementary oligonucleotide (complementary primer, Table 2) at a concentration of 10 µM to prevent the ribozyme from cleaving during transcription.

### 4.3. Kinetics of Cleavage

The purified *Bcep176* (C6) and (A6C), as well as uncleaved mouse gut HHRz RNA, were then incubated in the presence of different concentrations of Mg^2+^ or Mn^2+^ at 37 °C. The RNA from *Bcep176* (C6) was left to cleave for different times (2, 5, 10, 20 and 60 min) at 37 °C in the presence of 0.3, 1, 3 and 10 mM Mg^2+^ and 0.03, 0.1, 1 and 3 mM Mn^2+^. The uncleaved mouse gut HHRz was tested in the presence of 0.001, 0.003, 0.01, 0.03, 0.1 and 0.3 mM for both Mg^2+^ and Mn^2+^. Since the mutated mouse gut HHRz is less active, ten times greater concentrations were tested for both the cations. All the kinetic reactions were incubated in 25 mM KCl, 100 mM NaCl and 50 mM Tris-HCl (pH 7.5). For each time point, the reaction was stopped by using 2x gel loading buffer (10 mM EDTA (pH 8), 95% formamide, 0.05% (*w*/*v*) bromophenol blue and 0.05% (*w*/*v*) xylene cyanol).

### 4.4. Measure of the Cleavage and k_obs_

The time-course-stopped reactions were migrated on 8 M urea PAGE (8%) and exposed with phosphor imaging screens and scanned by Typhoon FLA9500. The sum of cleaved band intensities divided by the total (cleaved and uncleaved bands) was calculated. The data were plotted by using the formula of one-phase decay of GraphPad as below:
(1)$$y={\left(y_{0}-P\right)}^{-kX}+P$$
where *P* is plateau and *k* is the rate constant expressed in reciprocal of the X-axis time units. The plateau value (*P*) of the maximum corresponding divalent cation concentration was taken and used as a constraint for k_obs_ calculation because some reactions could have likely evolved for several additional hours. All the k_obs_ values were plotted against the concentrations of Mg^2+^ and Mn^2+^ for mouse gut HHRz (A6), mouse gut HHRz (A6C) and *Bcep176* (C6) (Appendix A).

## Figures and Tables

**Figure 1 ncrna-06-00014-f001:**
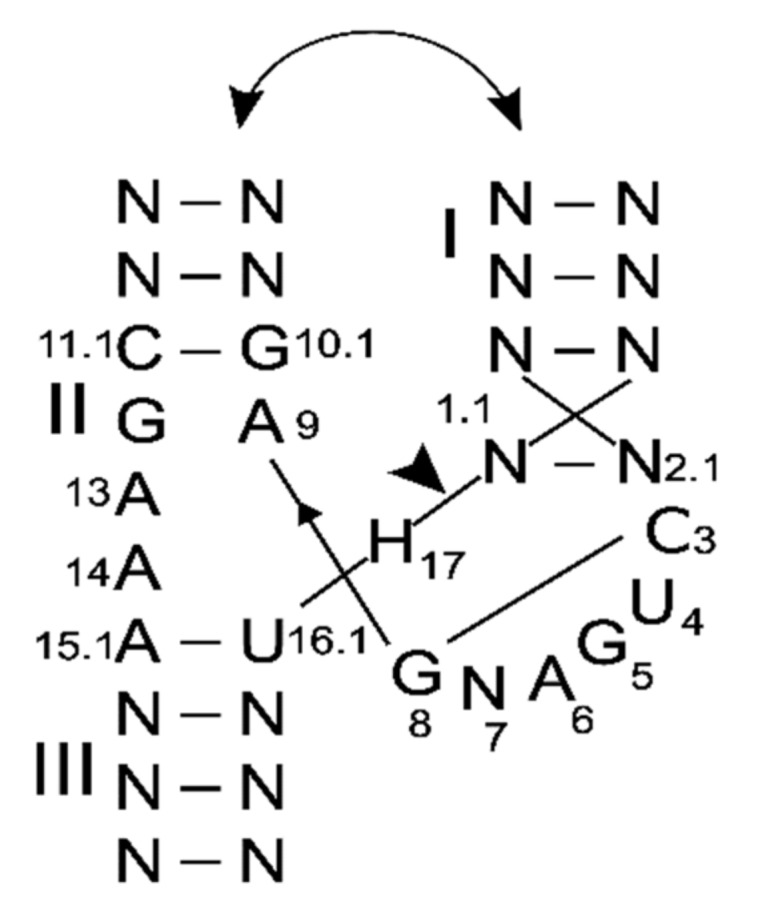
The structure and sequence consensus of hammerhead ribozymes (HHRz). The standard numbering of positions in the catalytic core of HHRz is shown. The cleavage site is indicated by an arrow. H: stands for all the nucleotides except G. The curved arrows illustrate the tertiary interaction between the stems I and II.

**Figure 2 ncrna-06-00014-f002:**
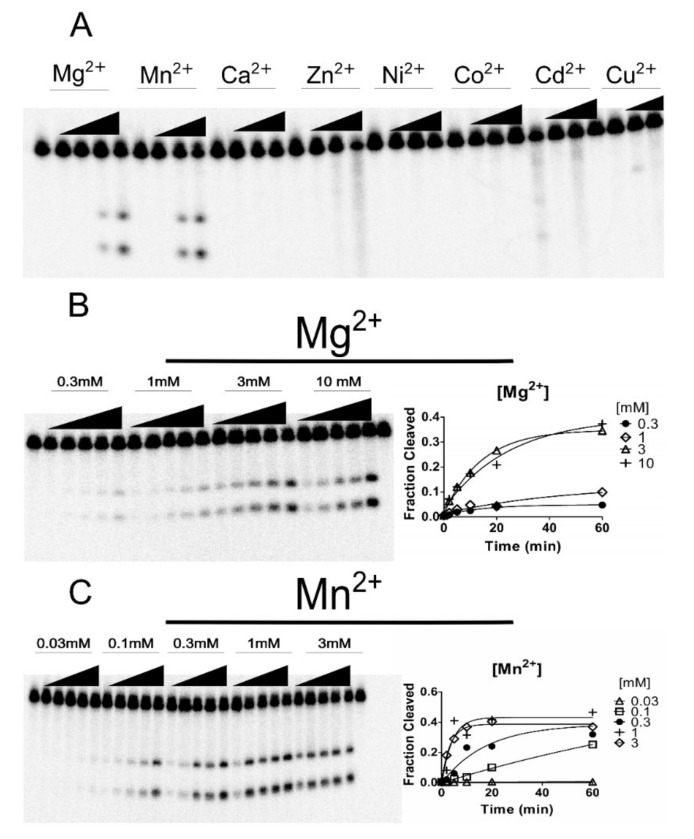
Cleavage assay for the hammerhead ribozyme *Bcep176* (C6). (**A**) Cleavage assay of *Bcep176* (C6) in the presence of other metals. For Mg^2+^, concentrations of 0, 0.01, 0.1, 1 and 10 mM were used. For the other metal ions tested, the concentrations were 0, 0.01, 0.1 and 1 mM. For assays with Cu^2+^, the concentrations were 0, 0.01 and 0.1 mM. (**B**,**C**) Cleavage assay of HHRz *Bcep176* (C6) in the presence of indicated concentrations of Mg^2+^ and Mn^2+^. Incubation times were 2, 5, 10, 20 and 60 min at 37 °C. The first and last lanes are negative controls (no divalent cations).

**Figure 3 ncrna-06-00014-f003:**
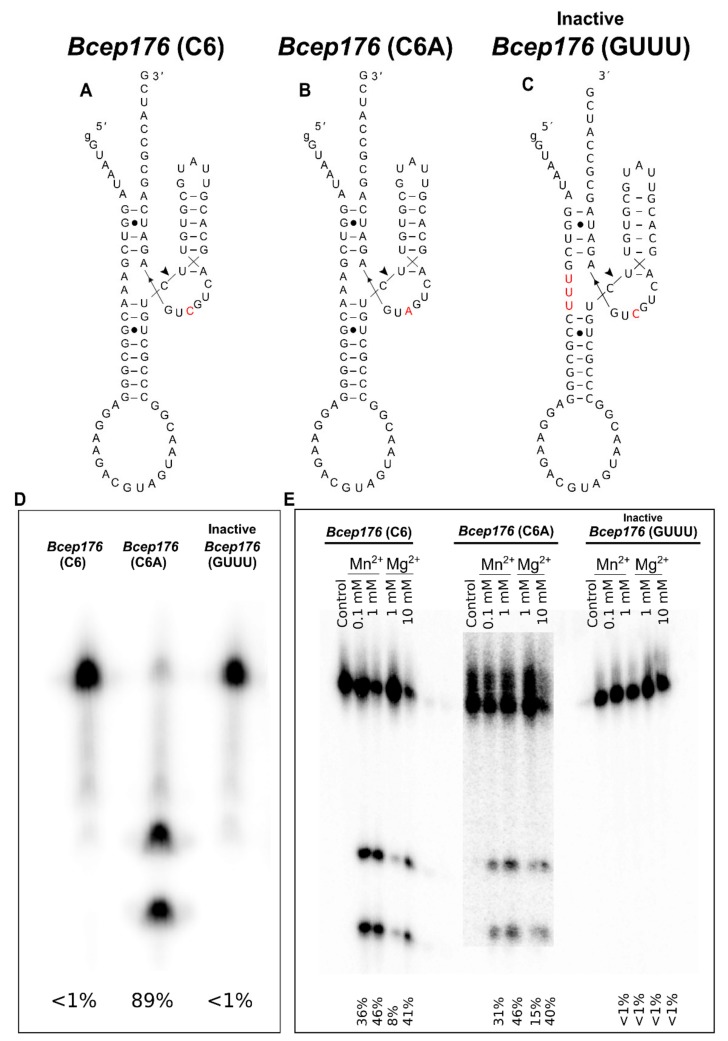
Importance of C6 in the bacteriophage hammerhead ribozyme *Bcep176*. (**A**) Native *Bcep176* (C6) bears a natural variation from the consensus catalytic core, C6, instead of A6, shown in red. (**B**) *Bcep176* (C6A) reversed to “standard consensus” (C6A), in red. (**C**) Introducing a mutation to change GAAA to GUUU in order to inactivate the ribozyme *Bcep176.* (**D**) Cleavage during in vitro transcription (in the presence of 25 mM Mg^2+^) for native *Bcep176* (C6), mutant *Bcep176* (C6A) and inactive *Bcep176* (GUUU). (**E**) Cleavage assay of *Bcep176* (C6), mutant *Bcep176* (C6A) and inactive *Bcep176* (GUUU) in the presence of 0.1 mM and 1 mM Mn^2+^ compared to 1 mM and 10 mM Mg^2+^. The incubation time was 60 min for all the assays.

**Figure 4 ncrna-06-00014-f004:**
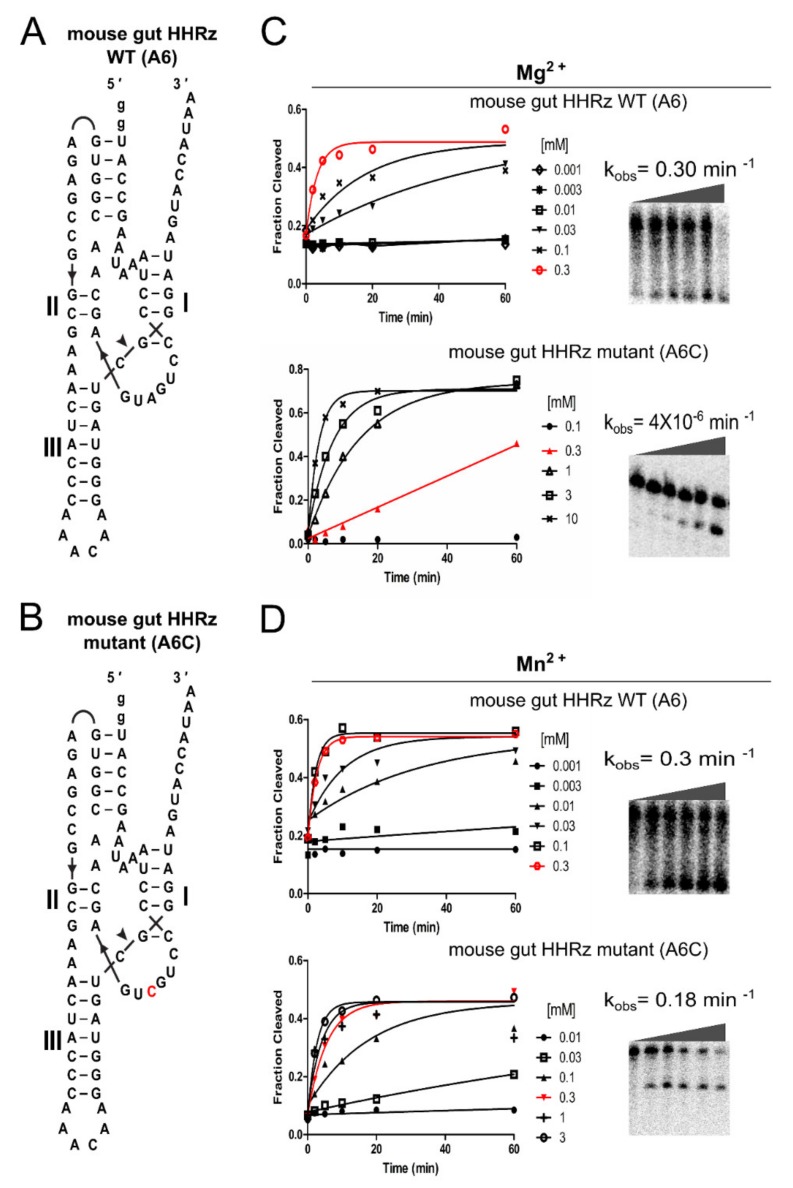
Effect of A6C mutation on self-cleavage activity for a pseudoknotted type II consensus core HHRz (A6) derived from the mouse gut metagenome (mouse gut HHRz). (**A**,**B**) Sequences and secondary structures of native mouse gut HHRz (A6) versus mutated mouse gut HHRz mutant (A6C), shown in red. (**C**,**D**) The gels are showing the self-cleavage activity of native mouse gut HHRz (A6) versus mutant HHRz (A6C) in the presence of 300 µM of Mg^2+^ or Mn^2+^. The graphs correspond to the fraction cleaved for all the concentrations indicated, red curves correspond to 300 µM. The incubation times were 0, 2, 5, 10, 20 and 60 min.

**Table 1 ncrna-06-00014-t001:** Cleavage rates of *Bcep176* (C6) variant and pseudoknotted type II mouse gut HHRz (mouse gut HHRz) with comparative Mn^2+^ and Mg^2+^ concentrations.

	*Bcep176*		Mouse Gut HHRz			
	WT(C6)		WT(A6)		Mutant(A6C)	
[mM]	Mg^2+^	Mn^2+^	Mg^2+^	Mn^2+^	Mg^2+^	Mn^2+^
0.001	ND	ND	ND	<10^−6^	ND	ND
0.003	ND	ND	1.2x10^−5^	0.0025	ND	ND
0.01	ND	ND	0.00070	0.031	ND	0.00093
0.03	ND	0.00012	0.019	0.096	ND	0.0070
0.1	ND	0.017	0.056	0.45	<10^−6^	0.056
0.3	0.0018	0.057	0.30	0.39	3.6 × 10^−6^	0.18
1	0.0041	0.31	ND	ND	0.067	0.24
3	0.051	0.29	ND	ND	0.14	0.37
10	0.041	ND	ND	ND	0.33	ND

The different k_obs_ obtained are presented for Mg^2+^ and Mn^2+^ (min^−1^). The red highlights show that at 0.3 mM concentration, Mn^2+^ is favored over Mg^2+^ for *Bcep176* (C6) and pseudoknotted type II HHRz derived from mouse gut (mouse gut HHRz) mutant (A6C), whereas green highlights no significant difference with Mn^2+^ vs Mg^2+^ for standard mouse gut HHRz (A6). ND stands for “not determined” for the indicated concentration.

**Table 2 ncrna-06-00014-t002:** Wild-type and mutant sequences for *Bcep176* and mouse gut HHRz and the primer sequences used.

Seq ID	Sequences
Bcep176 (C6)(Bcep176_Rev1 + Bcep176_Rev2)	gg AAUAGGUCGAAACGGCGGGAGGAAGACGUAGUAACGGCCCGCUGUCUGCACGUUAUGCGUGUACUGCUGAGAUCAGCGCCA
Bcep176 (C6A) (Bcep176_Rev2 + Bcep176_Rev4)	gg AAUAGGUCGAAACGGCGGGAGGAAGACGUAGUAACGGCCCGCUGUCUGCACGUUAUGCGUGUACUGaUGAGAUCAGCGCCA
Bcep176 (GUUU)(GAAA → GTTT) (Bcep176_Rev1 + Bcep176_Rev3)	gg AAUAGGUCguuuCGGCGGGAGGAAGACGUAGUAACGGCCCGCUGUCUGCACGUUAUGCGUGUACUGCUGAGAUCAGCGCCA
Bcep176_Rev1	TGGCGCTGATCTCAGCAGTACACGCATAACGTGCAGACAGCGGGCCGTTACTACGTCTT
Bcep176_Rev2	TAATACGACTCACTATAGGAATAGGTCGAAACGGCGGGAGGAAGACGTAGTAACGGCCC
Bcep176_Rev3	TAATACGACTCACTATAGGAATAGGTCGTTTCGGCGGGAGGAAGACGTAGTAACGGCCC
Bcep176_Rev4	TGGCGCTGATCTCAACAGTACACGCATAACGTGCAGACAGCGGGCCGTTACTACGTCTT
Rz mouse gut_HHRz (A6), (Rz mouse gut_fw + Rz mouse gut_rev)	gg UACCGAAUAAAUCCCCUGAUGAGCAACGGUGAGAGCCGGCGAAACUACCCAAACAAGGGUAGUCGGGAUAGUACCAUAA
Rz mouse gut_fw	TTCTAATACGACTCACTATAGGTACCGAATAAATCCCCTGaTGAGCAACGGTGAGAGCC
Rz mouse gut_rev	TTATGGTACTATCCCGACTACCCTTGTTTGGGTAGTTTCGCCGGCTCTCACCGTTGC
Rz mouse gut_HHRz (A6C), (Rz mouse gut_mutated_fw + Rz mouse gut_rev)	gg UACCGAAUAAAUCCCCUGcUGAGCAACGGUGAGAGCCGGCGAAACUACCCAAACAAGGGUAGUCGGGAUAGUACCAUAA
Rz_mouse gut_mutated_fw	TTCTAATACGACTCACTATAGGTACCGAATAAATCCCCTGcTGAGCAACGGTGAGAGCC
Complementary primer (to prevent cleavage during transcription)	GTAGTTTCGCCGGCTCTCACCGTTGCTCATCAGGGGATTTATTCGGTACC

Sequences of *Bcep176* (all versions) and mouse gut HHRz (WT A6) and mutant (A6C). Full sequences are underlined. The *Bcep176* (C6) and HHRz mouse gut HHRz (A6) were constructed from Bcep176_Rev1 + Bcep176_Rev2 and Rz mouse gut_fw + Rz mouse gut_rev, respectively. Two mutants for *Bcep176* (C6A and GUUU) as well as mouse gut HHRz mutant (A6C) were constructed by combining primers (Bcep176_Rev2 + Bcep176_Rev4) and (Bcep176_Rev1 + Bcep176_Rev3) for *Bcep176*, respectively, and (Rz_mouse gut_mutated_fw + Rz mouse gut_rev) for mouse gut HHRz. The red nucleotides indicate the mutation and the “gg” nucleotides in red at the start of the sequences are added for transcription.

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
