# Peer review of "Single Mutation in Hammerhead Ribozyme Favors Cleavage Activity with Manganese over Magnesium"

_ncrna, 2020, doi:10.3390/ncrna6010014_

Round 1
Reviewer 1 Report
Authors prepared just five hammerhead ribozymes (Bcep176 (C6), Bcep176 (C6A), Bcep176 (GUUU), mouse gut HHRz Type II (WT) and HHRz Type II (A6C) - others are not appeared in the manuscript) and analyzed their self-cleavage activity in the presence of several divalent metal ion. This revised manuscript is better than the last one, however, I cannot agree with the publication of this manuscript in ncRNA. I would like to make the following comments on the manuscript.
* The aim of this paper is unclear. While authors hypothesized that "some of these would be more likely to have varying ion specificity due to functional relevance to the genes they could potentially regulate or due to structural variations that might have been selected for such functions" (lines 61-63), there is no discussion about that on Bcep176 and mouse gut HHRz Type II. I recommend authors to rewrite the manuscript apart from the relationship with biological systems.
* All of the results about Bcep176 and its mutants are not important nor novel for all readers in ribozyme field.
* The large preferential difference between Mg2+ and Mn2+ seen in HHRz Type II (A6C) is the only important result of interest. Though the Mn2+ preference (actually weak activation with Mg2+) is interesting phenomenon, number of the examples are too limited to generalize the result. The description "...the impact of this mutation could be universal to HHRz" (lines 224-225) is overestimate if authors cannot explain the reason. A6 acts to interact with specified U for tighter folding and is not the direct coordination site of divalent cations. The reason for the metal preference of G12A mutant on the general base residue is totally different.
Minor (not so minor) points
* Unify the abbreviations in the figures, tables and text. For example,
Bcep 176, Bcep176 (space)
Bcep 176 (GUUU), GUUU, Inactive (Figure 3)
Bcep 176 (C6), WT C6,
mouse gut HHRz , HHRz Type II, consensus HHRz, Type II pseudoknotted HHRz, Type II consensus core HHRz, ... (Is it intend to confuse?)
25 mM Mg2+, 25 mM magnesium (ion), ... and so on.
* Regarding Figure 3
Add the information about incubation time.
Why WT C6 in panel D did not cleave in the presence of 25 mM Mg2+? In panel E, it cleaved itself 41% under 10 mM Mg2+.
* Regarding Figure 4
Time course of cleaved fraction about HHRz Type II (WT and A6C) should be shown in the figure.
Add time on panels B and C (or in the legend).
Results in panels B and C are very important but distorted and very messy with high BG. Replace to the proper figure.
Author Response
We thank the reviewers for their comments, as we think it helped us improve the manuscript significantly. Please find below a detailed answer to all of reviewer’s 1 comments (reviewers 2 and 3 accepted the manuscript). Comments of reviewer 1 are in bold below, followed by our answers. The complete manuscript with "track-changes" is also attached.
*The aim of this paper is unclear. While authors hypothesized that "some of these would be more likely to have varying ion specificity due to functional relevance to the genes they could potentially regulate or due to structural variations that might have been selected for such functions" (lines 61-63), there is no discussion about that on Bcep176 and mouse gut HHRz Type II. I recommend authors to rewrite the manuscript apart from the relationship with biological systems. All of the results about Bcep176 and its mutants are not important nor novel for all readers in ribozyme field.
#The main goal of this paper is focused to determine the catalytic activity of Bcep176 (C6) and to determine the effect of this variation on other hammerhead ribozyme, that is why we did an additional A6C mutation to a “standard” HHRz. As per the comments we had from the previous review, we tried to downplay the potential biological roles of this C6 variant, but still wanted to highlight the potential importance of it in Bcep176, with this naturally occurring core variation of one nucleotide which can favor one cation over others (at least in vitro). Nevertheless, we further down-played the biological roles by:
1- Deleting two introduction sentences describing the rationale of choosing ribozymes that may be involved in cation sensing so that we clearly introduce the main subject of the manuscript, the C6 variation. We kept only two short sentences at the beginning of the results section to very briefly explain that we looked at several ribozymes before finding this interesting case, which we chose for further study (otherwise we feel information is missing to explain why we decided to perform experiments on Bcep176).
2- We added this sentence to the discussion: “Nevertheless, we do not know if this is biologically relevant, especially given that concentrations of Mg2+ are typically a few orders of magnitude higher than for Mn2+.”
*The large preferential difference between Mg2+ and Mn2+ seen in HHRz Type II (A6C) is the only important result of interest. Though the Mn2+ preference (actually weak activation with Mg2+) is interesting phenomenon, number of the examples are too limited to generalize the result. The description "...the impact of this mutation could be universal to HHRz" (lines 224-225) is overestimate if authors cannot explain the reason.
#We have modified our sentence and deleted “…could be universal to all HHRz”.
#Also, as alluded to above, we kept results on Bcep176 because otherwise there is no apparent reason for us to start working on C6 mutants. Moreover, even if we do not have any in vivo data (as is the case for most HHRz papers), the fact that Bcep176 is a natural variant forces us to at least ask questions regarding the potential biological roles of this particular variant.
*A6 acts to interact with specified U for tighter folding and is not the direct coordination site of divalent cations. The reason for the metal preference of G12A mutant on the general base residue is totally different.
#According to some review (Nelson, J. A.; O. C. Uhlenbeck. Hammerhead redux: does the new structure fit the old biochemical data?. RNA 2008, 14, 605-615.) the A6 also interacts with G12 (in addition to U4, or rather the 2’OH of U4 ribose). Nevertheless, in all cases we agree that there is no data indicating that A6 interacts with a metal ion. Which is in accordance with what we wrote in the text.
# The case of G12A is indeed quite different, since it does interact with a metal ion. To make this clearer, we added the following text to the sentence mentioning G12:
“…, although in this case the reason for this change is better understood because this nucleotide directly interacts with the metal ion”
#Additional discussion added on the subject include:
“Among the tested cations, Cd2+ has coordination very similar to Mn2+ [42], which might thus be expected to affect the folding of ribozyme Bcep176 (C6) similarly as Mn2+, but this is not what cleavage activity suggests; implying the coordination is not the only factor affecting cleavage activity.”
Also, we amended the text in accordance to all the minor comments.
#The text and figures were uniformized for ribozyme denominations and their corresponding mutants and presented more clearly to eliminate confusion, as well as for metal ions (the atomic symbol with its charge was used everywhere except in the title, eg. Mg2+)
#The incubation times were added in the legend of figure 3
*Why WT C6 in panel D did not cleave in the presence of 25 mM Mg2+? In panel E, it cleaved itself 41% under 10 mM Mg2+.
#Answered in the text. Line 343-346:
“Note that the fact that Bcep176 (C6) barely cleaved during transcription (25 mM Mg2+) (Figure 3D), but cleaved to ~40% with 10 mM Mg2+ (Figure 2 and Figure 3E) may be due to differences of folding during in vitro transcription compared to folding after purification and snap cooling.”
*Regarding Figure 4:
*Time course of cleaved fraction about HHRz Type II (WT and A6C) should be shown in the figure.
*Add time on panels B and C (or in the legend).
*Results in panels B and C are very important but distorted and very messy with high BG. Replace to the proper figure.
#Significant changes were done to figure 4. All the graphs of time courses (previously in figure S1) were added to figure 4. The time is thus also shown on the graph.
#We have increased the resolution for some gels and “changed exposition” (by linearly changing contrast, to avoid inappropriate image manipulation) to reduce background. Also, we think that by adding the curves, thus showing the quantification of the cleavage, we can see that in spite of significant background, the gels do not provide erratic data and the data points fit relatively well with the curves. Moreover, we picked this concentration (0.3 mM) because at lower Mg2+ concentrations activity in the C6 ribozymes is almost non-existent (so a kobs could not be reliably calculated), but this also tends to confirm the trend we wanted to illustrate by comparing kobs of the ribozymes with each cation at 0.3 mM (i.e. that for C6 versions, a difference of a few orders of magnitude also likely exists between kobs-Mg and kobs-Mn at 0.1 mM concentrations).

Reviewer 2 Report
This paper discusses details of the structure and function of the Hammerhead ribozyme. I think this will be of interest to a fairly wide audience. The paper is clear and well presented. I find the review of past literature quite useful, and I also find the methods and experimental results section quite convincing - but I need to add that I am a theorist who is perhaps not an ideal reviewer for this paper. The paper contains lots of details of the effects of specific changes in the sequence. This appears relevant and interesting from the point of view of molecular evolution, but I am not close enough to the field to know just how original these things are and to what extent there have been similar findings by others.
In summary - I support publication of this paper with little or no modification, but I would recommend seeking a review from an experimentalist with closer knowledge about hammerhead ribozymes in particular.
Author Response
We thank the reviewer for the comments and we have done many changes to the text (mostly in response to reviewer 1 comments), this includes uniformizing abreviations and correcting many mistakes in the previous text.
Reviewer 3 Report
This study investigated the importance of single point mutation in modulating cation dependence of hammerhead ribozyme. THis ribozyme is known to be modulated by magnesium. in the present study, proper cleavage of the enzyme is observed with other cations as well. Because the catalytic core of the enzyme is highly conserved, preference of the enzyme towards specific cations is maintained. The present study determines that a naturally occurring variation in the catalytic core of the hammerhead ribozyme (A6C) can favor manganese over other cations.
Author Response
We thank the reviewer for the comments and we have done many changes to the text (mostly in response to reviewer 1 comments), this includes uniformizing abreviations and correcting many mistakes from the previous text.
Round 2
Reviewer 1 Report
I think the authors' response is still insufficient but acceptable. I agree with the publication of the paper in ncRNA.